# Comparison of Clinical Manifestation, Diagnosis, and Outcomes of Invasive Pulmonary Aspergillosis and Pulmonary Mucormycosis

**DOI:** 10.3390/microorganisms7110531

**Published:** 2019-11-05

**Authors:** Chun-Yu Lin, I-Ting Wang, Che-Chia Chang, Wei-Chun Lee, Wei-Lun Liu, Yu-Chen Huang, Ko-Wei Chang, Hung-Yu Huang, Hsuan-Ling Hsiao, Kuo-Chin Kao, Chung-Chi Huang, George Dimopoulos

**Affiliations:** 1Department of Pulmonary and Critical Care, Saint Paul’s Hospital, 330 Taoyuan, Taiwan; pitiful1984@gmail.com (C.-Y.L.); b9202071@cgmh.org.tw (H.-Y.H.); 2Department of Pulmonary and Critical Care, Chang Gung Memorial Hospital at Linkou, 333 Taoyuan, Taiwan; yuchenhahaha@gmail.com (Y.-C.H.); b9302072@cgmh.org.tw (K.-W.C.); kck0502@cgmh.org.tw (K.-C.K.); 3College of Medicine, Chang Gung University, 333 Taoyuan, Taiwan; 4Department of Pulmonary and Critical Care, Mackay Memorial Hospital, 10491 Taipei, Taiwan; Cherry.wang822@gmail.com; 5Department of Pulmonary and Critical Care, Chang Gung Memorial Hospital at Chiayi, 613 Chiayi, Taiwan; oceanpuma@gmail.com (C.-C.C.); wayneslow@gmail.com (W.-C.L.); 6School of Medicine, College of Medicine, Fu Jen Catholic University, 242 New Taipei City, Taiwan; medrpeterliu@gmail.com; 7Division of Critical Care Medicine, Department of Emergency and Critical Care Medicine, Fu Jen Catholic University Hospital, Fu Jen Catholic University, 243 New Taipei City, Taiwan; 8Department of Pharmacy, Chang Gung Memorial Hospital at Linkou, 333 Taoyuan, Taiwan; ellisa@adm.cgmh.org.tw; 9Department of Critical Care, ATTIKON University Hospital, University of Athens, Medical School, 12462 Athens, Greece; gdimop@med.uoa.gr

**Keywords:** outcomes, diagnosis, invasive pulmonary aspergillosis, invasive pulmonary mucormycosis

## Abstract

Objects: Invasive pulmonary mold infection usually has devastating outcomes. Timely differentiation between invasive pulmonary aspergillosis (IPA) from pulmonary mucormycosis (PM) is critical for treatment decision-making. However, information on IPA and PM differentiation is limited. Methods: We conducted a retrospective, multicenter, observational study, with proven and probable IPA and PM patients from January 2004 to December 2017. Demographics, clinical manifestations, image reports, histopathological findings, and outcomes were analyzed. Results: A total of 46 IPA (33 proven and 13 probable) and 19 PM (18 proven and one probable) cases were analyzed. The majority of tissues (81% in IPA and 61% in PM) were obtained using bronchoscopy. Prior influenza infection was a predisposing factor for IPA, and abscess formation in CT scan was associated with PM (*p* = 0.0491, *p* = 0.0454, respectively). The positive culture rate for PM was lower than that for IPA (37% vs. 67%, *p* = 0.0294). The galactomannan (GM) level from serum and bronchoalveolar lavage (BAL) fluid was significantly higher in IPA than in PM (3.3 ± 0.5 vs. 0.8 ± 0.6, *p* = 0.0361; 4.0 ± 0.6 vs. 0.59 ± 0.1, *p* = 0.0473, respectively). The overall mortality rate was 65%, which was similar among IPA and PM groups. Systemic steroid exposure and high Acute Physiology and Chronic Health Evaluation II (APACHE II) scores on admission were independently correlated to mortality in IPA (*p* = 0.027, *p* = 0.026, respectively). However, there was no predictor for mortality found in PM patients. Conclusions: Influenza infection, abscess formation in CT scan, and GM level may help physicians to differentiate IPA and PM. Bronchoscopy-guided biopsy and lavage specimen provide timely and definite diagnosis. The prognosis of IPA is associated with systemic steroid exposure and higher APACHE II scores on admission.

## 1. Introduction

The incidence of invasive pulmonary mold infection has increased rapidly in recent years and is not limited to immunocompromised patients [1,2,3]. Patients with influenza infection with a critically ill status, diabetes mellitus, chronic lung disease, chronic renal failure, and liver cirrhosis may also develop invasive pulmonary fungal infections [2,4,5,6]. *Aspergillus spp.* is the most common pathogen, followed by Mucorales [7]. Halo sign or air-crescent sign in chest computed tomography (CT) is the typical presentation in invasive pulmonary mold infection in neutropenic patients, but it has low sensitivity among non-neutropenic patients [8,9]. Both invasive pulmonary aspergillosis (IPA) and pulmonary mucormycosis (PM) have devastating outcomes; hence, early and timely distinguishing of IPA from PM are quite important for deciding the different treatment strategies [10]. However, accurate diagnosis is challenging for the identical presentations [1]. To our knowledge, there were only two studies discussing the differentiation of IPA from PM [10,11]. Chamilo et al. evaluated 29 IPA (eight had definite IPA, 21 had probable IPA) and 16 PM (nine had definite PM and seven had probable PM) patients with cancer and found that concomitant sinusitis and voriconazole prophylaxis were predictors for PM [10]. They also showed that multiple nodules and pleural effusion on initial CT scan were independently associated with PM [10]. Jung et al. studied 96 IPA (12 had proven IPA, 84 had probable IPA) and 24 PM (20 had definite PM and four had probable PM) patients, mainly with hematologic malignancy and who were receiving solid organ/stem cell transplantation, and found that reverse halo sign was more common in PM and airway invasiveness was the feature of IPA in CT scans [11]. However, the differentiation of IPA from PM in clinical manifestation and CT findings was not conclusive and the outcome analysis was lacking in relatively immuno-competent patients. The current study aimed to identify the differences in clinical manifestation, the diagnostic approach of IPA and PM, and to analyze their outcomes.

## 2. Results

### Patients’ Characteristics

This study included 46 IPA (33 proven IPA and 13 probable IPA) and 19 PM patients (18 with proven PM and one with probable PM, Figure 1). Among these patients, 19 IPA patients and eight PM patients were reported in our published article [4]. Thirty-one (67%) patients with IPA had a positive culture report from sputum, bronchial lavage fluid, or tissue, and only seven (37%) patients with PM had positive culture results (*p* = 0.0294, Figure 1 and Table 1). 

Table 1 demonstrated the comparison of baseline manifestations between IPA and PM. Age, sex, smoking history, underlying disease, sinusitis, and steroid exposure were not significantly different between groups. However, results in patients following influenza infection, who developed IPA, were independently higher than for PM patients (nine of 46, 20% vs. 0%, *p* = 0.0491). There were eight patients who developed breakthrough IPA and PM who were exposed to prior antifungal therapy empirically [12]. However, different prior antifungal agents were not correlated to either IPA nor PM. Thirty-nine (85%) IPA patients and 16 (84%) PM patients had received bronchoscopies and they had similar incidence of airway involvement (67% in IPA and 79% in PM, respectively, Table 1). In both IPA and PM, pseudomembranous form was the most frequent seen subtype (43% in IPA and 67% in PM), followed by ulcerative form (39% in IPA and 73% in PM) and obstructive form (15% in IPA and 13% in PM). There were 33 (72%) patients with proven IPA and 18 (95%) patients with proven PM. Others were probable cases. Histologic finding of IPA and PM are demonstrated in Figure 2. In these proven cases, the tissues were mainly obtained from bronchoscopies (81% in IPA and 61% in PM), followed by surgery and CT-guided biopsy. None of these patients developed massive hemoptysis or irreversible hypoxemia after procedures. Most of the fungal cultures were obtained using bronchoscopy in both IPA (81%) and PM (71%). However, the PM patients had a significantly higher negative culture rate versus IPA patients (12 of 19, 63% vs. 15 of 46, 33%, respectively, *p* = 0.0294). The level of Galactomannan (GM) from serum and bronchoalveolar lavage fluid (BAL) were significantly higher in IPA patients than in PM patients (3.3 ± 0.5 vs. 0.8 ± 0.6, *p* = 0.0361; 4.0 ± 0.6 vs. 0.59 ± 0.1, *p* = 0.0473; respectively). In patients who were exposed to a CT scan (74% in IPA and 89% in PM), consolidation was the most frequent finding in both group (82% in IPA and 94% in PM). There were four patients that had only airway involvement and all of them were IPA, but the difference was not significant versus PM patients (*p* = 0.2876). Comparing to IPA, PM was associated with more abscess formation (3 of 34, 9% vs. 6 of 17, 35%, respectively, *p* = 0.0454). The CT images were demonstrated in Figure 3. Patients with IPA had slightly higher APACHE II scores on admission (18.4 ± 1.4 vs.14.1 ± 1.8, respectively, *p* = 0.0804). There were 31 (67%) IPA patients and 13 (68%) PM patients who developed respiratory failure before diagnosis. More PM patients receiving Amphotericin B treatment at diagnosis (five, 26% vs. one, 2%, respectively, *p* = 0.0068). 

There were 42 deaths; 41 patients died from invasive fungal pneumonia, one had IPA with good response to voriconazole treatment, but died of brain stem ischemic stroke. Nonetheless, the in-hospital mortality because of invasive fungal pneumonia were similar in IPA and PM (28, 61% vs. 14, 74%, respectively, *p* = 0.4006). 

Predicting factors for in-hospital mortality are summarized in Table 2. Survival was not related to underlying disease in both IPA and PM. In univariate analysis, systemic steroid exposure, higher APACHE II score on admission, respiratory failure before diagnosis, and surgical intervention were significant predicting factors for in-hospital mortality in IPA. However, in multivariate analysis, only systemic steroid exposure and higher APACHE II score on admission were the independent prognostic factor in IPA (OR: 7.73, 95% CI: 1.8–33.1, *p* = 0.027; difference: 8.2, 95% CI: 2.95–13.47, *p* = 0.026, Table 2). In PM, no predictor for in-hospital mortality was found. 

## 3. Discussion

To the best of our knowledge, this report was the first one discussing the diagnostic approach and included the largest series of comparisons in clinical manifestations of patients with IPA and PM. Approximately half of our patients had DM and chronic lung disease. Only 10 patients (15%) had hematologic disease and seven of these were under neutropenic status. Influenza infection was predisposing to IPA. Abscess formation in CT scan provided a clinical hint for diagnosing PM. While PM had a lower culture rate in comparison to IPA, the GM level from serum and BAL were significantly higher among IPA patients. Seventy-two percent of patients with IPA and 95% of patients with PM were proven cases and two thirds of the diagnosing tissues were obtained from bronchoscopy. The mortality was greater than 60% in both IPA and PM patients. Systemic steroid exposure and APACHE II score on admission were independently correlated to mortality in IPA, but there was no predictor for in-hospital mortality among PM patients.

In recent decades, we found that not only immunocompromised patients were susceptible to fungal infections, patients with diabetes mellitus and chronic lung diseases may also experience invasive fungal infections [4]. In one review article, Patterson et al. demonstrated that critically ill patients who are admitted to intensive care units (ICUs) also have higher risk of pulmonary fungal infection [2]. Wauters et al. revealed that IPA is a more frequent complication in critically ill H1N1 patients and suggest that use of systemic steroid in these patients is an independent risk factor for fungal infections [5]. Moreover, in one large multicenter retrospective study, Schauwvlieghe et al. collected 457 critically ill patients with influenza, comparing to 321 patients with community acquired pneumonia, and demonstrated that influenza was independently associated with IPA [13]. In the current study, DM and chronic lung disease were the most common underlying diseases in both IPA and PM patients. The underlying diseases were similar in IPA and PM. However, we found that after influenza infection, patients were more susceptible to IPA rather than PM (20% vs. 0%, respectively, *p* = 0.0491). IPA is the most common mold infection, followed by PM [1]. Nevertheless, the incidence of PM seems to have risen in recent years [4,7,14]. Neofytos et al. found that after bronchoscopy initiation, the observed rate of invasive mucormycosis among hematopoietic stem cell transplantation (HSCT) recipients increased from 0.6% annually to 3% [7]. In our previous report, 25% of bronchoscopy-diagnosed invasive fungal tracheobronchitis were caused by Mucorales [4]. In the current investigation, among these invasive pulmonary mold-infected patients, one third of them had PM. Carol Garcia-Vidal et al. proposed that voriconazole prophylaxis was significantly associated with zygomycosis in hematologic patients [15]. Moreover, in animal models, the virulence of zygomycytes may increase after exposure to voriconazole [16]. In the current report, prior voriconazole exposure was slightly more frequently seen in PM but not statistically significant (11% in PM vs. 0% in IPA, *p* = 0.082, Table 1). Although the exposure to voriconazole may not directly lead to PM, the clinical awareness of PM should be risen when patients have had prior voriconazole exposure. 

Typical findings in CT scan, such as halo sign (HS) for IPA, reverse halo sign (RHS) for PM, and air-crescent sign, were useful in diagnosing invasive pulmonary mold infection in immunocompromised patients. In one review article, Georgiado et al. found lots of diseases, other than invasive pulmonary mold infection, presenting with those findings [8]. Nam et al. recently presented that consolidation or mass with halo sign were the most common findings on CT scan for PM. The serial morphologic changes into reverse halo sign, central necrotic cavity, or air-crescent sign were noted after treatment and recovery of neutropenia [17]. However, articles discussing about CT findings in non-neutropenic patients were scarce. Jung et al. conducted a retrospective study focusing on the comparison of CT scans for IPA and PM. They found that the RHS was more common in PM. But the RHS more frequently developed in neutropenic patients and half of them only appeared within five days from symptoms onset [11]. In our study, which was mainly composed of non-neutropenic patients, the most common finding on CT scan was consolidation (28 of 34, 82% in IPA vs. 16 of 17, 94% in PM, in patients who received CT scan, respectively). There were a total of seven patients (11%) who were under neutropenia status while diagnosing invasive mold infections, and four had consolidation in CT scan; one had merely airway invasion, one had cavitation, and one had abscess formation. None of our patients had HS or RHS. In the CT scan findings among our invasive pulmonary mold infection patients, only abscess formation was higher in PM patients than IPA (35% vs. 9%, respectively, *p* = 0.0454). 

The fungal culture takes a lot of time and positive rates in lower respiratory tract samples were 48% to 76% in IPA patients [11,18] and ranged from 29% to 46% in PM patients [11,19]. In the current report, the positive culture rate was significantly higher in IPA than in PM (67% vs. 37%, respectively, *p* = 0.0294). Among these patients, most of the positive culture specimens were obtained from bronchoscopy (81% in IPA and 71% in PM). GM is a fungal cell wall component that is released during tissue invasion by *Aspergillus* hyphae and can be detected in serum and bronchoalveolar lavage (BAL) fluid [20]. Serum levels in non-neutropenic patients may be underestimated because circulating neutrophils are able to clear the antigen [21]. Although the value of serum GM can be influenced by neutrophil counts [21,22], GM in BAL samples provides reliable diagnosing tools in IPA [20]. Moreover, GM had no role in diagnosing PM. In the current study, we demonstrated that GM in both serum and BAL fluid was significantly higher in IPA than in PM (3.3 ± 0.5 vs. 0.8 ± 0.6, *p* = 0.0361; 4.0 ± 0.6 vs. 0.59 ± 0.1, *p* = 0.0473; respectively). Considering the possible influence of neutrophil counts in serum GM, samples from BAL is a good choice in order to differentiate IPA from PM. Furthermore, in our research, 72% (33 of 46) of IPA and 95% (18 of 19) of PM were proven cases. Among them, 82% (27 of 33) of proven IPA and 61% (11 of 18) of proven PM tissue were obtained from bronchoscopy. Moreover, none of these patients developed massive hemoptysis or major complications after bronchoscopy. There were four patients that had concomitant IPA and PM during our chart review (data not included). Furthermore, Chamilos et al. also found five cancer patients that had concomitant IPA and PM [10]. Considering the concomitant IPA and PM, precise and timely diagnosis may still mainly depend on histopathology. In patients with a high risk of surgical intervention, biopsy from bronchoscopy and specimen from BAL for GM test and fungal culture may be an ideal diagnosing method. 

Many studies reveal that underlying conditions, such as neutropenia, active malignancy, liver disease, hematopoietic stem cell, or solid organ transplant, are risk factors for mortality in invasive mold infection [7,23,24,25]. In the current study, only 10 patients (15%) had hematologic disease and seven were under neutropenic status. We found that underlying disease had no association with in-hospital mortality in both IPA and PM. In addition to uncontrolled underlying disease, Taccone et al. showed that mechanical ventilation, renal replacement therapy during ICU stay, and higher sequential organ failure assessment score at diagnosis were independent predictors for death of invasive aspergillosis in critically ill patients [26]. Two thirds of our patients (67% in IPA and 68% in PM) developed respiratory failure before diagnosis and were admitted to ICU. The usage of systemic steroid, respiratory failure before diagnosis, and surgical intervention were correlated to mortality in IPA, but only systemic steroid exposure and high APACHE II score on admission were independent risk factors for in-hospital mortality among IPA patients. In regards to PM, Lin et al. conducted a retrospective study with more than 70% patients that had hematologic malignancy and were under neutropenic status. They demonstrated that concurrent bacteremia was the sole independent predictor for mortality among those PM patients [19]. However, in our PM patients, concurrent bacterial sepsis was not significantly correlated to outcome. Although the APACHE II score on admission was higher in the non-survivor group, the difference was not significant (15.9 ± 2.2 in non-survivor vs. 9.0 ± 1.8 in survivor, *p* = 0.0974, Table 2). These finding suggest that the outcomes of IPA and PM were more closely related to the clinical disease severity rather than underlying conditions. 

The present study has some limitations. First, for the retrospective nature of this study, the sample size was limited and some of the data are incomplete, this probably led to diminished generalizability; second, we only included proven and probable cases, patients with possible invasive mold infection may have been overlooked, and the outcomes may have been underestimated. 

## 4. Materials and Methods

### 4.1. Study Population

This retrospective, multicenter, observational study was conducted from January 2004 to December 2017 at the Linkou and Chiayi branches of Chang Gung Memorial Hospital, Far Eastern Memorial Hospital, and the Liouying branch of Chi Mei Medical Center. According to the revised definitions for invasive fungal infections from the European Organization for the Research and Treatment of Cancer/Mycosis Study Group (EORTC/MSG) [27], patients who met the criteria for proven or probable IPA or PM were included. 

### 4.2. Definitions and Classifications

Proven invasive fungal infection was diagnosed using histopathology with evidence of filamentous fungi and tissue invasion. Probable invasive fungal infection referred to the presence of a positive culture for fungal species or positive GM from BAL in specific hosts with classic image manifestations. In patients with fungal tracheobronchitis, the image findings were classified into pseudomembranous, ulcerative, and obstructive forms according to Denning’s classification [28]. 

Baseline characteristics, underlying conditions, laboratory parameters, image, and bronchoscopic findings, treatments, and outcomes were recorded. In-hospital mortality was accessed and the causes of death were also analyzed.

Informed consent was waived because this was a retrospective study and there was no modification in patient management. This study was approved by the institutional review boards of Chang Gung Memorial Hospital (CGMH 104-7452B). All personal information was encrypted in the database, and patient data accessed was de-identified. There was no breach of privacy.

### 4.3. Statistical Analysis

Categorical variables were described using counts (percentages) and continuous variables as means ± standard deviation (SD). We used Fisher’s exact test for categorical variables and the student’s *t* test for continuous variables. Independent predicting factors for in-hospital mortality were determined using multivariate logistic regression analysis. All analyses were two-sided, and *p* < 0.05 was considered statistically significant. Statistical analyses were performed using Prism version 5 (GraphPad Software Inc., La Jolla, CA, USA).

## 5. Conclusions

Differentiation for IPA and PM are difficult. Influenza infection and abscess formation in CT scan may help physicians to identify IPA from PM and initiate a different antifungal agent earlier. Bronchoscopy may be a safe and useful tool to obtain the specimen for fungal culture, GM test, and histopathology, and provide more precise and timely diagnosis. The outcome of IPA was associated with systemic steroid exposure and higher APACHE II score on admission. Concurrent bacterial sepsis was the only independent predictor for in-hospital mortality in PM. Further larger, prospective studies are required.

## Figures and Tables

**Figure 1 microorganisms-07-00531-f001:**
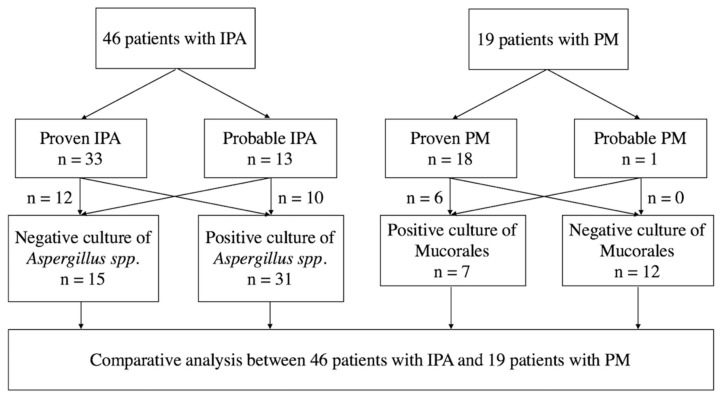
Schematic flow chart of the study. IPA, invasive pulmonary aspergillosis; PM, pulmonary mucormycosis.

**Figure 2 microorganisms-07-00531-f002:**
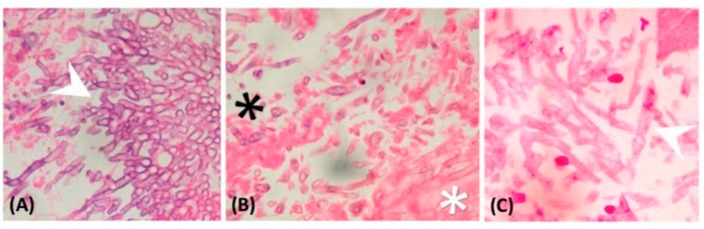
Histopathological finding: (**A**) Septate fungal hyphae branching at a 45° angle (arrowhead), which is characteristic of *Aspergillus spp.* (magnification: 400×); (**B**) broad-based, aseptate hyphae, which are characteristic of Mucormycete (white star), and the other septate fungal hyphae are *Aspergillus spp.* (black star) (magnification: 400×); (**C**) Mucormycete characterized with broad-based, aseptate hyphae (arrowhead) (magnification: 400×).

**Figure 3 microorganisms-07-00531-f003:**
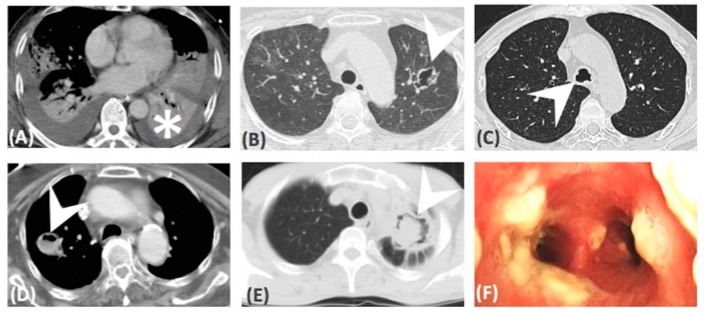
CT images: (**A**) Consolidation, (**B**) cavitation, (**D**) abscess formation, (**E**) ball in hole, (**C**,**F**) airway involvement only.

**Table 1 microorganisms-07-00531-t001:** Comparison of clinical characteristics, treatments, and outcomes between IPA and PM.

Variable	Univariate		
	IPA (*n* = 46)	PM (*n* = 19)	*p*-Value
Age, years, mean ± SD	61.2 ± 2.4	59.9 ± 3.9	0.7778
Sex, male, no. (%)	32 (70)	14 (74)	1.0
Smoking history, no. (%)	20 (43)	12 (63)	0.2863
Underlying disease, no. (%)			
DM	22 (48)	11 (58)	0.7893
Chronic lung disease	21 (46)	6 (32)	0.4082
Solid organ cancer	11 (24)	2 (11)	0.3146
Hematologic disease	5 (11)	5 (26)	0.1413
Neutropenia	3 (7)	4 (21)	0.1805
Cirrhosis	4 (9)	2 (11)	1.0
Organ transplantation	2 (4)	0 (0)	1.0
Autoimmune disease	4 (9)	0 (0)	0.3126
Sinusitis, no. (%)	5 (11)	5 (26)	0.2439
Systemic steroid, no. (%)	20 (43)	5 (26)	0.2656
Immunosuppressant, no. (%)	3 (7)	0 (0)	0.2326
Post influenza, no. (%)	9 (20)	0 (0)	0.0491
Prior antifungal agent, no. (%)			
Voriconazole	0 (0)	2 (11)	0.082
Amphotericin B	1 (2)	1 (5)	0.5024
Echinocandin	2 (4)	0 (0)	1.0
Breakthrough fungal infection	3 (7)	3 (16)	0.3469
Bronchoscopy, no. (%)	39 (85)	16 (84)	1.0
Airway involvement, no. (%)	31 (67)	15 (79)	0.5495
Scope pattern, no. (%)			
Pseudomembrane	20 (43)	10 (67)	1.0
Ulcerative	18 (39)	11 (73)	0.3525
Obstructive	7 (15)	2 (13)	0.6959
Diagnostic classification, no. (%)			0.0496
Proven	33 (72)	18 (95)	
Probable	13 (28)	1 (5)	
Proven cases, no. (%)			
Bronchoscopy	27 (81)	11 (61)	0.3483
Surgery	4 (12)	6 (33)	0.1562
CT-guided biopsy	3 (9)	1 (6)	1.0
Negative fungal culture, no. (%)	15 (33)	12 (63)	0.0294
Positive fungal culture, no. (%)	31 (67)	7 (37)	
Bronchoscopy	25 (81)	5 (71)	0.6236
Sputum	8 (26)	2 (29)	1.0
Tissue	1 (3)	0 (0)	1.0
GM			
Serum	3.3 ± 0.5	0.8 ± 0.6	0.0361
BAL	4.0 ± 0.6	0.59 ± 0.1	0.0473
CT finding, no. (%)	34 (74)	17 (89)	
Consolidation	28 (82)	16 (94)	0.4007
Cavitation	6 (18)	6 (35)	0.1811
Abscess formation	3 (9)	6 (35)	0.0454
Airway only	4 (12)	0 (0)	0.2876
Ball in hole	2 (6)	0 (0)	0.5467
APACHE II score on admission, mean ± SD	18.4 ± 1.4	14.1 ± 1.8	0.0804
RF before diagnosis, no. (%)	31 (67)	13 (68)	1.0
Concurrent bacterial sepsis, no. (%)	19 (41)	11 (58)	0.2786
Antifungal therapy at diagnosis, no. (%)			
Amphotericin B	1 (2)	5 (26)	0.0068
Voriconazole	18 (39)	3 (16)	0.085
Itraconazole	6 (13)	1 (5)	0.6633
Caspofungin	5 (11)	2 (11)	1.0
Surgical intervention, no. (%)	5 (11)	6 (32)	0.0670
In-hospital mortality of invasive mold pneumonia, no. (%)	28 (61)	14 (74)	0.4006

Abbreviation: SD, standard deviation; IPA, invasive pulmonary aspergillosis; PM, pulmonary mucormycosis; DM, diabetes mellitus; GM, Galactomannan; BAL, bronchoalveolar lavage fluid; CT, computer tomography; APACHE II, Acute Physiology and Chronic Health Evaluation II; RF, respiratory failure.

**Table 2 microorganisms-07-00531-t002:** Predicting factors for in-hospital mortality in IPA and PM.

Variable	IPA				PM			
	Univariate			Multivariate	Univariate			Multivariate
	Survivor *n* = 18	Non-Survivor *n* = 28	*p*-Value	*p*-Value	Survivor *n* = 5	Non-Survivor *n* = 14	*p*-Value	*p*-Value
Systemic steroid, no. (%)	3 (17)	17 (61)	0.0055	0.027	0 (0)	5 (36)	0.2565	
Post influenza, no. (%)	2 (11)	6 (21)	0.4525		0 (0)	0 (0)	1.0	
Neutropenia, no. (%)	1 (6)	2 (7)	1.0		1 (20)	2 (14)	1.0	
APACHE II score on admission, mean ± SD	13.2 ± 1.6	21.4 ± 1.7	0.0030	0.026	9.0 ± 1.8	15.9 ± 2.2	0.0974	0.178
RF before diagnosis, no. (%)	6 (33)	25 (89)	0.0002	0.461	2 (40)	11 (79)	0.2621	
Concurrent bacterial sepsis, no. (%)	5 (28)	14 (50)	0.2199		1 (20)	10 (71)	0.1108	0.124
Airway involvement, no.(%)	10 (56)	21 (75)	0.4964		4 (80)	11 (79)	1.0	
Antifungal therapy at diagnosis, no. (%)			0.1326				0.2261	
Amphotericin B	0 (0)	1 (4)			2 (40)	3 (21)		
Voriconazole	7 (39)	9 (32)			0 (0)	4 (29)		
Oral itraconazole	5 (28)	1 (4)			1 (20)	0 (0)		
Caspofungin	1 (6)	4 (14)			1 (20)	1 (7)		
Surgical intervention, no.(%)	5 (28)	0 (0)	0.0063	0.99	3 (60)	3 (21)	0.2621	

Abbreviation: IPA, invasive pulmonary aspergillosis; PM, pulmonary mucormycosis; APACHE II, Acute Physiology and Chronic Health Evaluation II; RF, respiratory failure.

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
