# Peer review of "Comparison of Clinical Manifestation, Diagnosis, and Outcomes of Invasive Pulmonary Aspergillosis and Pulmonary Mucormycosis"

_microorganisms, 2019, doi:10.3390/microorganisms7110531_

Round 1

Reviewer 1 Report

This study compares clinical presentation and risk factors between 47 cases with IPA and 22 cases with pulmonary mucormyoses.

The study has potential but requires MAJOR revisions nd quite some work to become suitable for publication.

Specific comments:

A total of 8 cases had prior antifungal drugs. These 8 cases need to be classified into whether they meet criteria of breakthrough infection or not (use the new criteria from the ECMM andMSG https://onlinelibrary.wiley.com/doi/full/10.1111/myc.12960). Breakthrough infection should then also be included as a variable into Table 1 and potentially the multivariable model.

Is your model really a multivariate model or in fact a multivariable model? See https://www.ncbi.nlm.nih.gov/pmc/articles/PMC3518362/

Is a multivarible analysis really appropriate for such small n? At the very least results of univariate logistic regression analysis and multivariable logistic regression analyses should be displayed properly with ORs and aORs plus 95%CIs and p-value. ALso the approach needs to be described in much more detail and finally the validity and predictive power of the final multivariable model needs to be evaluated by calculating Hosmer-Lemeshow and AUC. This accounts for both Table 1 and Table 2, where the logistic regression analysis needs to be removed and extra tables added for these analyses.

The dataset would provide a great opportunity to also add a variable on quality of care into Table 1 by calculating the EQUAL Mucor and EQUAL Aspergillus scores for all cases. I would strongly recommend that. See https://www.ncbi.nlm.nih.gov/pubmed/30770712

https://www.ncbi.nlm.nih.gov/pubmed/29944740

Grammar: Please be more careful. Currently authors switch around between present and past (e.g. "is made" in methods)

Table 2: What is a "Ployene"?? Give details on the formulation and add what dosages were used

Author Response

Response to Reviewer 1 Comment:

Point 1:

This study compares clinical presentation and risk factors between 47 cases with IPA and 22 cases with pulmonary mucormyoses.

The study has potential but requires MAJOR revisions nd quite some work to become suitable for publication.

Specific comments:

A total of 8 cases had prior antifungal drugs. These 8 cases need to be classified into whether they meet criteria of breakthrough infection or not (use the new criteria from the ECMM andMSG https://onlinelibrary.wiley.com/doi/full/10.1111/myc.12960). Breakthrough infection should then also be included as a variable into Table 1 and potentially the multivariable model.

Response 1:

Thanks for the great and kind reminder. All the 8 cases were breakthrough infection according to the new criteria from MSG and ECMM. They use antifungal therapy empirically, because of the critical illness and leukemia histories. After modified the data according to Reviewer 2’s suggestion, there were 6 patients had breakthrough infection. We had mentioned the breakthrough infection in the result.

Point 2:

Is your model really a multivariate model or in fact a multivariable model? See https://www.ncbi.nlm.nih.gov/pmc/articles/PMC3518362/

Is a multivarible analysis really appropriate for such small n? At the very least results of univariate logistic regression analysis and multivariable logistic regression analyses should be displayed properly with ORs and aORs plus 95%CIs and p-value. ALso the approach needs to be described in much more detail and finally the validity and predictive power of the final multivariable model needs to be evaluated by calculating Hosmer-Lemeshow and AUC. This accounts for both Table 1 and Table 2, where the logistic regression analysis needs to be removed and extra tables added for these analyses.

Response 2:

Thanks for your kind reminder. We use multivariate analysis for these data. After removed 4 patients with concomitant IPA and PM, we didn’t do further multivariate analysis for Table 1. For Table 2, we had added OR and 95% CI in results.

Point 3:

The dataset would provide a great opportunity to also add a variable on quality of care into Table 1 by calculating the EQUAL Mucor and EQUAL Aspergillus scores for all cases. I would strongly recommend that. See https://www.ncbi.nlm.nih.gov/pubmed/30770712

https://www.ncbi.nlm.nih.gov/pubmed/29944740

Response 3:

Thanks for your friendly reminder. The two scoring systems are great and quite important in clinical care for PM and IPA. However, limited to our retrospective design and some patients were diagnosed several years ago (no TDM evaluation, nor drug sensitivity test, or GM test), we couldn’t provide more information using these scoring systems. We would try to care patients with the new scoring system.

Point 4:

Grammar: Please be more careful. Currently authors switch around between present and past (e.g. "is made" in methods)

Response 4:

Yes, that’s our mistake. We had corrected as possible. We also had sent this article to English editing and the certification had been attached. Please let us know if you find any problems.

Point 5:

Table 2: What is a "Ployene"?? Give details on the formulation and add what dosages were used

Response 5:

Thanks for correction. We had specified to Amphotericin B. The dose was standard dose as 50 mg per day.

Reviewer 2 Report

The addressed question is of high interest

Major comments:

Patients with coinfection with PM and IPA should be excluded from the study as the histopathology findings cannot exclude coinfection. Those patients should not be reclassified in IPA or PM.

PM patients with positive GM Ag in BAL or serum should be excluded from PM for comparison with IA; there should be considered as coinfected

How patients with PM and negative culture were confirmed to have PM?

Previously reported criteria associated to PM or IPA should be studied: pleural effusion, multiple nodule, sinus involvement?

How many cases were disseminated?

Minor comments:

Table 1 form should be improved

Change Bronchoscopyc to bronchoscopy

Ployene to polyene

Chamilo to Chamilos

Bronchoscopy is repeted in table 1

Was there patients with RHS?

Ball in hole: is it still invasive mold infection?

Why vori or itra were used in PM cases?

As all patients from tracheobronchitis study were included in probably biaised this recruitment

Author Response

Response to Reviewer 2 Comment:

Point 1:

Patients with coinfection with PM and IPA should be excluded from the study as the histopathology findings cannot exclude coinfection. Those patients should not be reclassified in IPA or PM.

Response 1:

Yes, Thanks for the great opinion. We had re-analyzed these data after removing 4 patients with concomitant IPA and PM.

Point 2:

PM patients with positive GM Ag in BAL or serum should be excluded from PM for comparison with IA; there should be considered as coinfected

Response 2:

Yes, Thanks for the great opinion. We had re-analyzed these data after removing 4 patients with concomitant IPA and PM.

Point 3:

How patients with PM and negative culture were confirmed to have PM?

Response 3:

In our 22 PM cases, 21 had significant Mucorales tissue invasion base on histopathology. Only one patient was probable case, and the culture was positive for Mucorales.

Point 4:

Previously reported criteria associated to PM or IPA should be studied: pleural effusion, multiple nodule, sinus involvement?

How many cases were disseminated?

Response 4:

Only 4 IPA patients had pleural effusion study and proven as aspergillus related empyema. We only included pulmonary invasive infection and none in our study was disseminated case.

Point 5:

Table 1 form should be improved

Change Bronchoscopyc to bronchoscopy

Ployene to polyene

Chamilo to Chamilos

Bronchoscopy is repeted in table 1

Was there patients with RHS?

Ball in hole: is it still invasive mold infection?

Response 5:

We had modified Table 1 as your kind reminder.

In Regarding to Bronchoscopy in Table 1, the first one mentioned the overall patients who receiving bronchoscopy and the second one means that in proven cases, who had been diagnosed by bronchoscopy.

None of these patients had RHS.

Ball in hole was observed in 3 patients, who had also demonstrated pulmonary consolidation and clinical sepsis. Moreover, the lesion progressed after CT scan as the infection process. Thus, we also presented these cases CT reports.

Point 6:

Why vori or itra were used in PM cases?

Response 6:

In these 6 cases, they were treated as empiric therapy, but the medication was modified after the diagnose known.

Point 7:

As all patients from tracheobronchitis study were included in probably biaised this recruitment

Response 7:

Our previously reported article was discussing about invasive fungal tracheobronchitis in mechanically ventilated critically ill patients. They were all invasive infection and all of them had parenchyma involvement with high mortality. That’s the reason why we included these cases as invasive fungal infection. Moreover, based on the previous experience, we found that it’s quite difficult in differentiation Aspergillus from Mucorales, even we had seen the airway involvement. Thus, we conducted this study aims to figure out the clinical presentation for early differentiation.

Round 2

Reviewer 1 Report

Thank you for trying to implement my suggestions which has improved the manuscript.

As a note EQUAL scores could still be calculated even if the care is less than perfect and e.g. no TDM was performed. These scores would then be less then perfect and also allow for better interpretation to the observed mortality rates.

Also please do careful proofreading of the manuscript (still find the term ployene in one of the tables).

Author Response

Response to Reviewer 1 Comment:

Point 1:

As a note EQUAL scores could still be calculated even if the care is less than perfect and e.g. no TDM was performed. These scores would then be less then perfect and also allow for better interpretation to the observed mortality rates.

Response 1:

Thanks for the great and kind reminder. We tried to calculate the score and found the scoring was around 4 to 6 in both IPA and PM. First, our patients mainly composed with non-immunocompromised patients (Only 10 had hematologic disease and 7 of them developed neutropenia), only few patients met the neutropenia criteria. Moreover, we had no PCR test, susceptibility testing, nor TDM. We also had not molecular diagnostic tool. None had follow up CT scan after treatment. Thus, it’s quite difficult to interpret the EQUAL score among our retrospectively collected data. But our hospital had developed TDM and susceptibility testing recently. We will care patients with the new scoring system in the future.

Point 2:

Also please do careful proofreading of the manuscript (still find the term ployene in one of the tables).

Response 2:

Please accept my sincere apology. We had corrected it.